# Correlation between Post-Acute Electroconvulsive Therapy Alpha-Band Spectrum Power Increase and Improvement of Psychiatric Symptoms

**DOI:** 10.3390/jpm11121315

**Published:** 2021-12-06

**Authors:** Hideyuki Iwanaga, Takefumi Ueno, Naoya Oribe, Manabu Hashimoto, Jun Nishimura, Naho Nakayama, Nami Haraguchi, Hiroshi Tateishi, Yutaka Kunitake, Yoshito Mizoguchi, Akira Monji

**Affiliations:** 1Department of Psychiatry, National Hospital Organization, Hizen Psychiatric Medical Center, 160 Mitsu, Yoshinogari, Kanzaki, Saga 842-0192, Japan; hideiwan@gmail.com (H.I.); n.oribe@mac.com (N.O.); ma9hashi@yahoo.co.jp (M.H.); kurokikuma@gmail.com (J.N.); 2Division of Clinical Research, National Hospital Organization, Hizen Psychiatric Medical Center, 160 Mitsu, Yoshinogari, Kanzaki, Saga 842-0192, Japan; naho.nakayama.h259@gmail.com (N.N.); harunami79@gmail.com (N.H.); 3Department of Neuropsychiatry, Graduate School of Medical Sciences, Kyushu University, 3-1-1 Maidashi, Higashi-ku, Fukuoka 812-8582, Japan; 4Department of Psychiatry, Faculty of Medicine, Saga University, 5-1-1 Nabeshima, Saga 849-8501, Japan; tateishh@cc.saga-u.ac.jp (H.T.); kunitake@cc.saga-u.ac.jp (Y.K.); ymizo@cc.saga-u.ac.jp (Y.M.); amonji@hf.rim.or.jp (A.M.)

**Keywords:** brief psychiatric rating scale, electroconvulsive therapy, mood disorders, quantitative electroencephalogram, schizophrenia

## Abstract

The results of quantitative electroencephalography (qEEG) studies on electroconvulsive therapy (ECT) have been inconsistent, and indicators of the efficacy of ECT have not been clearly identified. In this study, we examined whether qEEG could be used as an indicator of the effect of ECT by measuring it during the course of treatment. We analyzed qEEG data before and after acute-phase ECT in 18 patients with schizophrenia, mood disorders, and other psychiatric disorders. We processed the qEEG data and compared the spectral power between the data acquired before and after ECT. The spectral power increased significantly after ECT in the delta, theta, and alpha bands. There was a strong significant correlation between the increase in the spectral power of the alpha band after acute ECT and improvement in the Brief Psychiatric Rating Scale score. Our results suggest that an increase in the alpha-band spectral power may be useful as an objective indicator of the treatment effect of acute ECT.

## 1. Introduction

Electroconvulsive therapy (ECT), first performed in 1938 [1], has since been used for the treatment of schizophrenia and mood disorders. Despite the development of psychotropic drugs as effective treatments for various psychiatric disorders, ECT has retained an important clinical role. ECT was initially used for the treatment of schizophrenia, but this use has been reduced with the advent of antipsychotics. However, a recent review has shown that it is not only beneficial as an augmentation strategy for treatment-resistant schizophrenia, but is also effective in a variety of other schizophrenic conditions [2]. In addition to schizophrenia, it has been reported to be effective for the treatment of several psychiatric disorders including bipolar disorder and depression [3]. Despite having indications for a variety of psychiatric disorders, the treatment course of ECT, which is indicated for both schizophrenia and mood disorders, has very rarely been analyzed in the literature.

Various studies have investigated the mechanisms of action of ECT and attempted to identify factors that can predict its efficacy. Thus far, the presence of catatonia symptoms and a short duration of psychiatric disorders have been established as predictive factors of high efficacy [4,5]. Two recent meta-analyses have demonstrated that ECT affects peripheral brain-derived neurotrophic factors, closely related to the pathology of psychiatric diseases [6,7]. Recent reports have also shown that inflammatory processes are related to ECT’s mechanism of action as well as to factors predicting its increased efficacy [8,9]. 

Quantitative electroencephalography (qEEG) is a relatively simple method more easily performed in clinical settings than other electrophysiological and functional brain imaging methods [10,11]. Patients with good responses to ECT have exhibited hyperactivity (higher theta activity) in the rostral anterior cingulate cortex in pretreatment EEG, and follow-up analyses have demonstrated the specificity of this finding, which was not confounded by age or pretreatment depression severity [12]. A report on the use of ECT for the treatment of psychotic depression demonstrated a positive association between an increase in theta waves in the subgenual anterior cingulate gyrus and the improvement in psychotic symptoms after ECT; the degree of the decrease in theta waves in the same region before ECT was shown to predict the antipsychotic response to ECT [13]. One study emphasized that high-frequency gamma-band oscillations are frequently altered in patients with psychiatric disorders, that baseline gamma can act as a surrogate for pharmacodynamics in predicting the time course of clinical treatments, and that alterations in evoked (time-locked) gamma power may serve as a disease biomarker that can be used to assess patient responses to new treatments [14]. Existing ECT neuroimaging data varies considerably between studies, and therefore has not yet allowed for the formulation of a unified hypothesis for the mechanism of action of ECT [15]. To clarify this, we analyzed qEEG data including high-frequency band activity during ECT treatment for psychiatric symptoms in patients with severe psychiatric disorders. In this study, we used qEEG analysis including high-frequency band, which has previously been difficult to measure, and the Brief Psychiatric Rating Scale (BPRS), which can evaluate both schizophrenia and mood disorders simultaneously, to evaluate the clinical course.

## 2. Materials and Methods 

### 2.1. Participants

We recruited 25 patients with schizophrenia, mood disorders, and other psychiatric disorders diagnosed using the Diagnostic and Statistical Manual of Mental Disorders, Fifth Edition [16]. Patients were judged by multiple experienced psychiatrists and anesthesiologists to be eligible for ECT therapy, and ECT application was considered safe in all cases. Patients were evaluated using BPRS-24, and we explored the relationship between psychiatric symptoms and qEEG during the course of treatment. Patients who agreed to participate in this study were enrolled at the National Hospital Organization Hizen Psychiatric Medical Center between 1 November 2017 and 10 January 2020. The selection criteria were based on the American Psychiatric Association Guidelines [17], and were as follows: patients with schizophrenia, mood disorders, and/or other psychiatric disorders deemed eligible for safe ECT by multiple psychiatrists and anesthesiologists and who had agreed to participate in this study. The exclusion criterion was a physical condition that increased the risk of ECT and made ECT infeasible after discussion among psychiatrists and anesthesiologists. Physical conditions that increase the risk of ECT include intracranial occupying lesions, increased intracranial pressure, recent myocardial infarction, recent intracerebral hemorrhage, unstable aneurysms/vascular malformations, pheochromocytoma, and a high risk of anesthesia (indicated by a score of 4 or 3 by the Society of Anesthesiology standards). Data from a total of 18 participants were finally included in the analysis. Some data had to be excluded from the analysis as the EEG used in this study was not a research EEG, and some data were significantly affected by electromyographic contamination (*n* = 4) or electrode instability (*n* = 3). In addition, some patients were unwilling to undergo EEG examinations before ECT. Patient demographic characteristics are presented in Table 1. The study was conducted according to the guidelines of the Declaration of Helsinki and approved by the Institutional Review Board (or Ethics Committee) of Saga University (protocol code 2017-06-05 and date of approval 4 September 2017). Informed consent was obtained from all subjects involved in the study. 

### 2.2. ECT Therapy

Acute ECT was performed twice weekly according to the American Psychiatric Association Guidelines [17]. A pulse-wave therapy device (Thymatron System IV; Somatics LLC, Lake Bluff, IL, USA) was used as the energizing device, and only in one case was it changed to a sine-wave therapy device (CS-1 type; Sakai Medical Device, Tokyo, Japan). The electrodes had a bilateral frontal arrangement (with a pulse width of 0.5 ms) or right unilateral arrangement (with a pulse width of 0.25 ms). In the bilateral frontal arrangement, the energy was initially applied according to the half-age strategy and increased or decreased according to the seizure waveform (high amplitude and continuity), presence or absence of postictal suppression, and degree of improvement in clinical symptoms. In the right unilateral arrangement, titration was performed to confirm the first seizure threshold; the second energization was performed with an energy 2.5 to 6 times the seizure threshold and then increased or decreased according to the seizure waveform (amplitude height and continuity). The energy was also increased or decreased depending on the presence or absence of postictal suppression and degree of improvement in clinical symptoms. For the bilateral frontal arrangement, even if the energy was set to 100% (the largest in Japan) and energization was performed, the seizure waveform could be poor, postictal suppression was not always observed, and/or clinical symptoms did not always improve. In such cases, the pulse width was increased to 0.75, 1.00, and 1.25 ms. Even with this widening of the pulse width, one case showed no clinical improvements, and a sine-wave therapy device was used instead. The applied voltage of the sine-wave therapy device was 100 V, the first energization was performed at 5 s, and the voltage was increased by 5 V depending on the seizure waveform (high amplitude and continuity), presence or absence of postictal suppression, and degree of improvement in clinical symptoms; the energization time was increased to 10 s. There were no drug treatment changes during ECT therapy. Acute-phase ECT was completed when doctors, nurses, and family members determined that the patient’s condition had improved before the appearance of the symptoms targeted for treatment. Psychiatric symptoms were assessed using BPRS-24 before and after ECT therapy (Figure 1) [18]. BPRS-24 is one of the most frequently used instruments by clinicians and researchers to measure psychiatric symptoms [19]. It comprises 24 items that assess the severity of psychiatric symptoms. In this study, we chose BPRS-24 to keep the assessments short and simple and to ensure inter-rater reliability. In addition, BPRS-24 can also assess a wider range of diseases than other scales, which is advantageous since the subjects who underwent ECT did not have a uniform diagnosis, and because the status evaluation (e.g., catatonia) was more important than the disease. BPRS-24 was assessed by junior psychiatrists (one to three years of experience) pretrained and audited by senior doctors.

The anesthesia administered during ECT was as follows: remifentanil (1 µg/kg/min div) and propofol (0.5 kg), which were used as intravenous anesthetics. Rocuronium (0.6 kg) was used as a muscle relaxant, and propofol and rocuronium were increased or decreased as appropriate. Oxygen was administered and artificial respiration performed after intravenous injection of a muscle relaxant. Anesthesia management was performed by an anesthesiologist. Blood pressure, electrocardiogram, blood oxygen saturation, anesthetic depth, EEG, and electromyogram were continuously monitored during the procedure.

### 2.3. EEG Recordings

All 25 participants underwent EEG recordings before and two weeks after ECT (Figure 1) using the 19 scalp locations according to the standard 10–20 system of electrode placement (i.e., Fp1, Fp2, F7, F3, Fz, F4, F8, T7, C3, Cz, C4, T8, P7, P3, Pz, P4, P8, O1, O2). EEG was acquired using a digital EEG instrument (EEG-1224; Nihon-Kohden, Tokyo, Japan) while the subjects lay quietly with their eyes closed in a sound-attenuated room. Ag/AgCl electrodes were used, and impedance was maintained at <5 KΩ. EEG data were recorded at a 16-bit resolution, a sampling rate of 500 Hz, and a time constant of 0.3 s, and low-pass filtered at 120 Hz and notch-filtered at 60 Hz. EEG data were recorded using an apparatus commonly used in clinical practice and included eye-open/closed, photic stimulation, and hyperventilation. The stored EEG data were processed offline using the Brain Vision Analyzer package (Brain Products, Gilching, Germany). Data were filtered with a 0.5 Hz high-pass filter, a 200 Hz low-pass filter, and a 60 Hz notch filter. For each subject, 60 s of EEG data before photic stimulation was manually selected and segmented into 2 s epochs. Artifact rejection was performed manually based on visual inspection. Segments containing blinking artifacts, muscle or cardiac contamination, or any other massive artifacts were excluded. Next, epochs with signals >±100 µV were excluded. Subjects with <20 segments (40 s) at either time point were excluded from further analysis. As a result, data from a final total of 18 participants (eleven males and seven females) were used in the subsequent analysis. Corrected EEG data were transformed into current source density estimates using a spherical Laplacian algorithm, and fast Fourier transforms were conducted using a Hanning window with a 10% taper length. The mean absolute power was computed for each of the following frequency bands: delta (0.5–4 Hz), theta (4–8 Hz), alpha 1 (8–10 Hz), alpha 2 (10–12 Hz), beta 1 (12–20 Hz), beta 2 (20–30 Hz), gamma 1 (30–58 Hz), and gamma 2 (62–100 Hz). Each participant’s data were averaged across epochs for each electrode. The spectral power was averaged for the electrodes selected according to a topographic map (Figure 2).

### 2.4. Statistical Analysis

Statistical analyses were performed using SPSS 27.0.0 (IBM Corp., Armonk, NY, USA). Paired *t*-tests were used to compare variables between time points. A Spearman’s ρ (two-tailed) was used in the exploratory analysis of the correlations between changes in each frequency range of interest and changes in the symptom score. The significance level for all analyses was set at *p* < 0.05. The BPRS-24 change score was calculated as follows:100 × [{value at Time 1 − value at Time 2}/value at Time 1].(1)

Note that the statistical conclusions reported remained the same when we excluded a case who was treated with a sine wave.

## 3. Results

### 3.1. Spectral Power

Delta, theta, and alpha (1 and 2) power were significantly increased after ECT (delta: t = −4.4, *p* = 0.001, Cohen’s d = −1.04; theta: t = −3.9, *p* = 0.001, Cohen’s d = −0.91; alpha 1: t = −2.8, *p* = 0.01, Cohen’s d = −0.65; alpha 2: t = −3.0, *p* = 0.008, Cohen’s d = −0.70). We found no significant change in beta or gamma power after ECT (−1.3 < t < 1.8, 0.09 < *p* < 0.374, −0.30 < Cohen’s d < 0.43; Figure 3).

### 3.2. Correlation between Spectral Power and Clinical Variables

The BPRS total score significantly decreased after ECT (*n* = 18, t = 7.4, *p* < 0.001, Cohen’s d = 1.74). Note that this improvement in BPRS held true when calculated for all participants (*n* = 25, t = 8.8, *p* < 0.001, Cohen’s d = 1.79), suggesting that the exclusion of some participants due to excessive artifacts may not have had a significant impact on the results of the statistical analysis. For the frequency ranges significantly different before vs. after ECT, exploratory analysis of the correlations between the percent change in spectral power and the percent change in BPRS score was performed using Spearman’s ρ. For alpha 1 and alpha 2, a greater increase in spectral power after ECT was significantly correlated with a greater decrease in BPRS score (alpha 1: rho = −0.65, *p* = 0.04; alpha 2: rho = −0.67, *p* = 0.003; Figure 4). 

## 4. Discussion

We investigated qEEG before and after ECT in a clinical setting. We found that the spectral power significantly increased after ECT in the delta, theta, and alpha ranges. Furthermore, the improvement in psychotic symptoms measured using BPRS-24 significantly correlated with an increase in the power spectrum in the alpha range. We predicted that changes in gamma band activity would have the greatest impact on symptom changes, but found no such correlation. The same was true for the beta band. This may indicate that comparing the spectral power of the alpha band before and after acute-phase ECT would be sufficient to evaluate the degree of clinical improvement in pathological conditions requiring acute-phase ECT such as schizophrenia.

In terms of psychiatric treatments that complement pharmacotherapy, evidence for transcranial magnetic stimulation (TMS) for depression has been accumulating in recent years. However, its indications are still limited, and electroconvulsive therapy is still an important complementary treatment for schizophrenia and various other drug-resistant psychiatric disorders such as those of the participants in this study. 

In patients with schizophrenia, Chen et al. [20] reported a robust increase in slow waves (delta and theta waves in particular) and alpha waves in the frontal lobe. Zhao et al. [21] recorded EEG before and after acute ECT in 26 patients with schizophrenia who were taking clozapine, observing a decrease in alpha waves and an increase in theta waves in the frontotemporal region after ECT. Inconsistencies in alpha activity before and after ECT may be attributable to differences in the timing of qEEG analysis and the heterogeneity of diagnoses. 

In patients with depression, Bruder et al. [22] observed higher pretreatment alpha power in selective serotonin reuptake inhibitor responders. Compared with healthy controls, lower alpha activity has also been reported in patients with euthymic bipolar disorder [23]. Conversely, increased right alpha synchronization has been reported in patients with schizophrenia-like psychosis and epilepsy with psychotic symptoms [24]. Thus, in psychiatric illnesses, the alpha-band spectral power appears to differ between patients with and without psychosis and could be used to predict treatment responsiveness. An explanation of the present study’s results is that EEG was measured two weeks after the end of acute ECT. We performed continuous ECT after acute phase ECT and measured the clinical effects. Continuous ECT was initiated two weeks after the end of acute phase ECT; thus, EEG measurements were performed in advance. This was unavoidable, as this was a clinic-based study. 

We found a strong correlation between the increase in alpha-band spectral power after acute ECT and BPRS score improvement. Therefore, the correlation between the rate of change in alpha-band power and BPRS total score is most likely due to the improvement in mood symptoms. However, ECT contributes to the improvement in overall brain function [25], not just mood symptoms, which may manifest as a normalization of alpha waves.

Our results suggest that focusing on alpha-band spectral power may be helpful for predicting the therapeutic response to ECT, though the EEG findings measured before treatment alone did not correlate with symptom changes. The present study also shows that EEG measurements may provide an objective measure to indicate the response to ECT treatment for psychiatric disorders. Future studies with a larger sample size should be conducted to clarify the relationship between alpha band activity and the effects of ECT. 

The limitations of this study should be noted. In this study, we followed the clinical course and EEG of patients who were eligible for ECT regardless of the diagnosis. It would be desirable to increase the number of cases and study the relationship with diagnosis in the future. In addition, it was difficult to analyze the high-frequency EEG because of the very fine noise from electrophysiological detection. With the development of future equipment, we hope to be able to study the relationship between ECT and physiology including high-frequency EEG in more detail. 

## Figures and Tables

**Figure 1 jpm-11-01315-f001:**
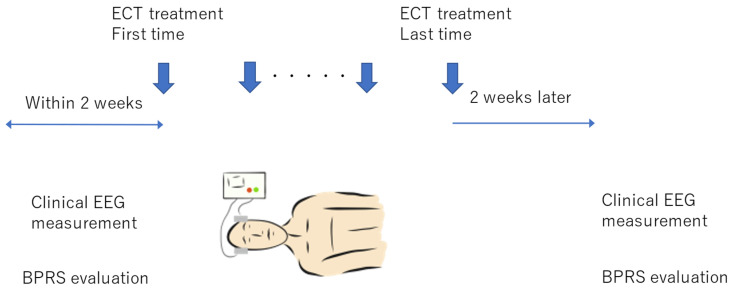
Study flowchart showing the sequence and timing of electroconvulsive therapy (ECT), electroencephalography (EEG), and the Brief Psychiatric Rating Scale (BPRS).

**Figure 2 jpm-11-01315-f002:**
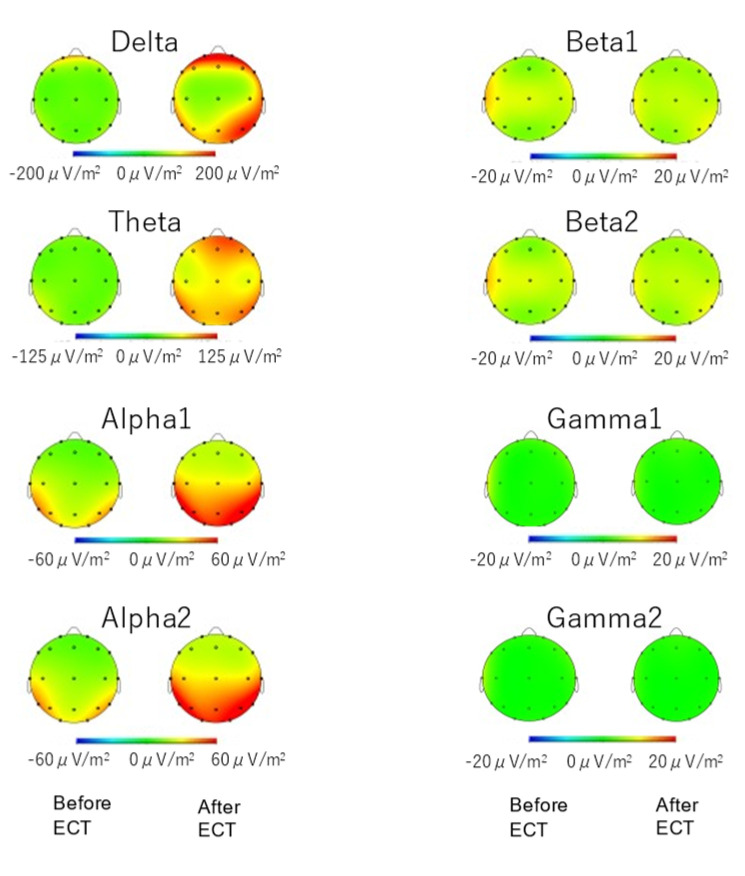
Grand average topographical plot of electroencephalography power in delta, theta, alpha 1, alpha 2, beta 1, beta 2, gamma 1, and gamma 2 for both time points (before and after electroconvulsive therapy (ECT)). In alpha 1 and alpha 2, the electrodes O1, O2, P3, P4, T5, T6, and Pz were selected to average the spectral power. In other ranges, all electrodes were selected to average the power.

**Figure 3 jpm-11-01315-f003:**
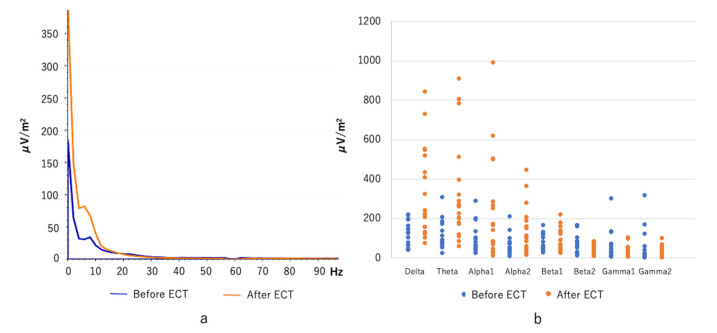
Averaged spectra plots (**a**) and individual power values (**b**) in delta (0.5–4 Hz), theta (4–8 Hz), alpha 1 (8–10 Hz), alpha 2 (10–12 Hz), beta 1 (12–20 Hz), beta 2 (20–30 Hz), gamma 1 (30–58 Hz), and gamma 2 (62–100 Hz) for both time points (before and after electroconvulsive therapy (ECT)). Spectral power was significantly increased at delta, theta, alpha 1, and alpha 2.

**Figure 4 jpm-11-01315-f004:**
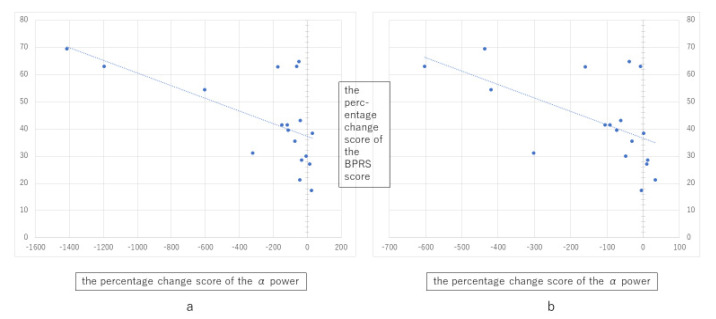
Scatterplots showing the relationship between percent change in BPRS total score and (**a**) percent change in alpha 1 power and (**b**) percent change in alpha 2 power. The X-axis is the percentage change score in alpha power and the Y-axis is the percentage change score in BPRS score.

**Table 1 jpm-11-01315-t001:** Demographic characteristics.

No.	Sex	Age (y)	Diagnosis	Number of ECT Treatments	Pulse-Wave Therapy Device			Sine-Wave Therapy Device		BPRS		Onset Age (y)	Recent Illness Period (mo)	Illness Episodes (mo)	Concomitant Psychotropics
					EA	E_max_ (%)	PW_max_ (msec)	V_max_ (V)	ET (sec)	Before	After				
1	M	27	Sc	12	bf	40	0.5			86	52	20	3	2	HPD9mg, Que400mg
2	M	26	Sc	15	bf	90	0.5			70	41	18	9	2	Ola20mg, HPD9mg
3	M	32	Sc	12	bf	15	0.5			90	62	25	13	3	Ris12mg, Que200mg
4	M	19	Sc	12	bf	15	0.5			98	81	15	21	3	Clo600mg
5	M	30	Sc	27	bf	100	0.5			105	75	24	27	3	Zot300mg
6	F	70	MD	7	bf	35	0.5			43	26	52	3	8	Mir45mg, Ola5mg
7	M	44	Sc	12	bf	100	0.75			61	48	16	14	15	Ris12mg, Ase20mg
8	F	45	Sc	19	bf	100	1			121	37	22	1	5	Ola5mg
9	M	57	BD	26	bf	100	1.25	115	10	50	35	39	21	5	Mil50mg
10	F	68	MD	12	bf	100	0.5			113	42	67	3	2	Esc20mg, Ris8mg
11	F	61	MD	13	bf	45	0.5			78	48	60	14	1	Ven225mg, Mir15mg, Ola20mg
12	M	86	MD	12	ru	60	0.25			65	38	85	4	2	Dul40mg, Mir15mg, Tra50mg
13	M	70	MD	15	ru	70	0.25			71	25	69	9	1	Esc20mg, Mir45mg, Ari3mg
14	F	63	BD	10	ru	50	0.25			48	31	50	6	6	VPA 600mg, Que112.5mg
15	M	69	Sc	12	ru	50	0.25			90	41	30	6	2	Ase15mg, CPZ100mg
16	F	47	ATPD	15	bf	50	0.5			92	50	47	1	1	Que300mg
17	F	67	MD	9	ru	50	0.25			73	35	66	4	1	Esc20mg, Que300mg, Ris1mg
18	M	50	Sc	15	bf	100	0.5			74	54	19	60	10	Clo450mg
Avg		51.7								79.3	45.6	40.2	12.2	4	
SD		19.1								22	15.3	22.2	14.1	3.7	

Abbreviations: Ari, aripiprazole; Ase, asenapine; ATPD, acute transient psychotic disorder; Avg, average; BD, bipolar depression; bf, bilateral frontal; BPRS, Brief Psychiatric Rating Scale; Clo, clozapine; CPZ, chlorpromazine; Dul, duloxetine; EA, electrode arrangement; E_max_, maximum amount of energy; Esc, escitalopram; ET, energizing time; F, female; HPD, haloperidol; M, male; MD, major depression; Mil, milnacipran; Mir, mirtazapine; mo, months; Ola, olanzapine; PW_max_, maximum pulse width; Que, quetiapine; ris, risperidone; ru, right unilateral; Sc, schizophrenia; SD, standard deviation; Tra, tradozone; Ven, venlafaxine; V_max_, Maximum voltage VPA, valproic acid; y, years; Zot, zotepine.

## Data Availability

The data that support the findings of this study are available on request from the corresponding authors upon reasonable request. The data are not publicly available due to privacy or ethical restrictions.

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
