# Peer review of "Correlation between Post-Acute Electroconvulsive Therapy Alpha-Band Spectrum Power Increase and Improvement of Psychiatric Symptoms"

_jpm, 2021, doi:10.3390/jpm11121315_

Round 1

Reviewer 1 Report

Manuscript ID: jpm-1478428

Title: Correlation Between Post-Acute Electroconvulsive Therapy Alpha-Band

Spectrum Power Increase and Improvement of Psychiatric Symptoms

Authors: Hideyuki Iwanaga, Takefumi Ueno *, Naoya Oribe, Manabu Hashimoto,

Jun Nishimura, Naho Nakayama, Nami Haraguchi, Hiroshi Tateishi, Yutaka

Kunitake, Yoshito Mizoguchi, Akira Monji

Toward the Establishment of Personalized Medicine in Psychiatry

Short summary

The authors investigate spectral EEG in psychiatric patients before and after ECT treatment and report significantly increased spectral power in the delta to alpha bands in EEG, as well as improvement of symptoms according to a clinical questionnaire.

Comments

The manuscript is very well written, and I can recommend publication in the Journal of Personalized Medicine after addressing some minor issues:

  1. In the abstract, the authors suggest their results of alpha-band spectral power could predict therapeutic response to ECT. I think this result may be missing from the results section, as only the change of EEG before/after and the correlation of EEG with the questionnaire is reported. I would love to see the additional analyses/results that enable prediction of response to ECT. I would have expected machine learning, or at least an analysis at least like “alpha 1 band power before ECT correlated to BPRS score after ECT”. Maybe the authors tried to answer this question with the analyses in 3.2 and Figure 4, but 1) these are introduced as exploratory, and 2) it should be explained why percentage change rates were chosen to answer this question. Importantly, it seems to me like based on these findings and given a patient’s before ECT EEG spectral data, one could not predict the ECT response.
  2. The reported p-values and Figure 3 are impressive. I think additional reporting of effect sizes for all findings would be worthwhile.
  3. There may be a mistake in line 164, where the authors speak of a high-pass filter at 120 Hz, which would be very uncommon in EEG. I guess this was the low-pass filter. However, why was the low-pass filter during preprocessing (line 168/169) set higher than the recorded one, to 200 Hz?
  4. The authors should please state explicitly the number of electrodes used for the recording, and also mention which reference was used.
  5. The number of ECT session and the patients’ age covered a wide range. It seems interesting to me to consider these as covariates in exploratory analyses.

Reviewer 2 Report

The manuscript by Iwanaga et al. addresses very important question and presents a set of unique data.

I suggest the following improvements:

Would the results change if a person following sine wave stimulation is excluded?

Do authors find any associations to the age or number of ECT procedure?

Please provide average spectra plots before and after stimulation.

Figure 2: font is too small 

Figure 3: what is plotted? SD? SE? Not electroencephalography but electroencephalogram

Figure 4: too overlapping, some data points are difficult to see due to the overlap with text

Phrasing: It is better to use change score or percent change  than “percent change rate”. 

Authors should consider discussion results from the point of TMS stimulation as well to complement findings in medication therapy. 

Round 2

Reviewer 2 Report

Figure 3. Authors state that they present spectra plots. That is not the case they seem to have added individual power values. I suggest adding Spectral plots and keeping the individual values (part B of the figure) instead of part A.

Author Response

Reviewer2-1: Figure 3. Authors state that they present spectra plots. That is not the case they seem to have added individual power values. I suggest adding Spectral plots and keeping the individual values (part B of the figure) instead of part A.

We apologize for misinterpreting the suggestions from the reviewer. As suggested, we presented the averaged spectral plot as Figure 3a and the individual power as Figure 3b. We edited the figure legend. “Figure 3. Averaged spectra plots (a) and individual power values (b) in delta (0.5–4 Hz), theta (4–8 Hz), alpha 1 (8–10 Hz), alpha 2 (10–12 Hz), beta 1 (12–20 Hz), beta 2 (20–30 Hz), gamma 1 (30–58 Hz), and gamma 2 (62–100 Hz) for both time points (before and after electroconvulsive therapy [ECT]).  Spectral power was significantly increased at delta, theta, alpha 1, and alpha 2.”